# Thymoquinone Ameliorates Carfilzomib-Induced Renal Impairment by Modulating Oxidative Stress Markers, Inflammatory/Apoptotic Mediators, and Augmenting Nrf2 in Rats

**DOI:** 10.3390/ijms241310621

**Published:** 2023-06-25

**Authors:** Marwa M. Qadri, Mohammad Firoz Alam, Zenat A. Khired, Reem O. Alaqi, Amani A. Khardali, Moudi M. Alasmari, Ahmad S. S. Alrashah, Hisham M. A. Muzafar, Abdullah M. Qahl

**Affiliations:** 1Department of Pharmacology and Toxicology, College of Pharmacy, Jazan University, Jazan 45142, Saudi Arabia; 2Inflammation Pharmacology and Drug Discovery Unit, Medical Research Center (MRC), Jazan University, Jazan 45142, Saudi Arabia; 3Surgical Department, Faculty of Medicine, Jazan University, Jazan 45142, Saudi Arabia; 4Department of Clinical Pharmacy, College of Pharmacy, Jazan University, Jazan 45142, Saudi Arabia; 5College of Medicine, King Saud Bin Abdulaziz University for Health Sciences, Jeddah 22384, Saudi Arabia; 6King Abdullah International Medical Research Center (KAIMRC), Jeddah 22384, Saudi Arabia; 7Pharmacy Administration, Ministry of Health, Health Affairs General Directorate, Najran 66251, Saudi Arabia

**Keywords:** carfilzomib, thymoquinone, nephrotoxicity, nephroprotective, antioxidant, anti-inflammatory, gene expression

## Abstract

Chemotherapy-induced kidney damage is an emerging problem that restricts cancer treatment effectiveness. The proteasome inhibitor carfilzomib (CFZ) is primarily used to treat multiple myeloma and has been associated with severe renal injury in humans. CFZ-induced nephrotoxicity remains an unmet medical need, and there is an urgent need to find and develop a nephroprotective and antioxidant therapy for this condition. Thymoquinone (TQ) is a bioactive compound that has been isolated from Nigella sativa seeds. It has a wide range of pharmacological properties. Therefore, this experimental design aimed to study the effectiveness of TQ against CFZ-induced renal toxicity in rats. The first group of rats was a normal control (CNT); the second group received CFZ (4 mg/kg b.w.); the third and fourth groups received TQ (10 and 20 mg/kg b.w.) 2 h before receiving CFZ; the fifth group received only TQ (20 mg/kg b.w.). This experiment was conducted for 16 days, and at the end of the experiment, blood samples and kidney tissue were collected for biochemical assays. The results indicated that administration of CFZ significantly enhanced serum marker levels such as BUN, creatinine, and uric acid in the CFZ group. Similarly, it was also noticed that CFZ administration induced oxidative stress by reducing antioxidants (GSH) and antioxidant enzymes (CAT and SOD) and increasing lipid peroxidation. CFZ treatment also enhanced the expression of IL-1β, IL-6, and TNF-α production. Moreover, CFZ increased caspase-3 concentrations and reduced Nrf2 expression in the CFZ-administered group. However, treatment with 10 and 20 mg/kg TQ significantly decreased serum markers and increased antioxidant enzymes. TQ treatment considerably reduced IL-1β, IL-6, TNF-α, and caspase-3 concentrations. Overall, this biochemical estimation was also supported by histopathological outcomes. This study revealed that TQ administration significantly mitigated the negative effects of CFZ treatment on Nrf2 expression. Thus, it indicates that TQ may have utility as a potential drug to prevent CFZ-induced nephrotoxicity in the future.

## 1. Introduction

The kidney is the main route of antineoplastic agent excretion. Chemotherapy-induced nephrotoxicity is a significant complication that decreases the efficacy of cancer treatment. Drug-associated nephrotoxicity affects many components of the nephron structure and, clinically, can manifest in electrolyte disturbances or an increase in acute kidney markers [1]. A proteasome inhibitor (Carfilzomib) is used for treating several myelomas and has been associated with severe kidney failure in humans [2,3,4,5].

Carfilzomib (CFZ) is widely disseminated to all body tissues except the blood–brain barrier after intravenous administration [6]. CFZ is largely metabolized by plasma and tissue-resident peptidases and epoxide hydrolases rather than by the liver. [6]. CFZ-based chemotherapy has demonstrated survival benefits for relapsed and refractory myeloma (RRMM), but it has also been associated with severe renal adverse effects [4,5]. Clinical studies showed that CFZ-treated patients present with acute renal injury due to thrombotic microangiopathy and tumor lysis syndrome (TLS) [7,8]. According to a case report, 25% of CFZ-treated patients experience renal vasoconstriction-related insults, which can be avoided with N-acetyl-L-cysteine, but the rest of the consequences are still unknown [9]. Hence, more research is required to examine a potential nephroprotective drug against CFZ-mediated kidney damage and discover the underlying molecular mechanism.

Efentakis et al., 2022, previously established a CFZ-induced nephrotoxicity animal model to assess the molecular mechanism of CFZ-mediated renal injury. Efentakis et al., 2022, showed that CFZ causes nephrotoxicity by activating serum/glucocorticoids-regulated kinase 1 (SGK-1)/mineralocorticoid receptors in the kidneys of mice [10]. Additionally, Efentakis et al., 2022, thoroughly investigated the molecular mechanisms implicated in CFZ-induced nephrotoxicity, including apoptosis, reactive oxygen species (ROS), and inflammation-related pathways [10]. The study showed that 8 mg/kg of CFZ increased serum creatinine and serum urea-bound nitrogen at 24 h compared to the control [10]. Another study conducted by Al Harbi et al., 2019, showed that 4 mg/kg of CFZ increased ROS levels and inflammation via the NF-κB signaling pathway in Wistar albino rats [11]. The NRF2 signaling pathway plays a crucial function in protecting cells from oxidative injury caused by elevated levels of reactive oxygen species (ROS). ROS-activated NRF2 induces the expression of antioxidant enzymes, including heme oxygenase, catalase, glutathione peroxidase, and superoxide dismutase, which neutralize ROS and protect cells from oxidative stress damage. Activation of NRF2 signaling in cancer cells, however, results in chemoresistance by inactivating drug-mediated oxidative stress and protecting cancer cells from drug-induced cell death [12,13]. According to Al Harbi et al.’s 2019 study, CFZ at a dose of 4 mg/kg increased caspase-3 in rats [11]. Conclusively, CFZ-associated nephrotoxicity remains an unmet medical need, and there is a significant need to identify and discover a nephroprotective and antioxidant therapy against CFZ-induced nephrotoxicity.

The main active ingredient in *N. sativa* seeds is thymoquinone (TQ), which has antiproliferative, anti-inflammatory, antioxidant, anticancer, and anti-fibrotic effects [14,15,16]. Previous research has found that TQ treatment reduced hyperglycemia in experimental type 2 diabetes, LPS-induced renal fibrosis, and doxorubicin-induced cardiotoxicity in male rats [17,18]. Recently, Al Fayi et al., 2020, showed that TQ treatment protects against cisplatin-induced kidney injury by ameliorating Nrf2/HO-1-mediated signaling [19]. Taking all evidence together, in this study, we attempted to determine whether TQ treatment would reduce CFZ-associated rat nephrotoxicity.

## 2. Results

### 2.1. Impact of TQ Treatment on Kidney Function Markers

CFZ administration (4 mg/kg) resulted in a significant (*p* < 0.0001) increase in kidney function markers such as blood urea nitrogen (139%), uric acid (279%), and creatinine (216%) compared to the control. TQ treatment (10 mg/kg) resulted in a significant decrease in BUN (−24.47%; *p* < 0.0001), creatinine (−69.64%; *p* < 0.0001), and uric acid (−15.19%; *p* < 0.001) compared to the CFZ group. Similarly, the highest doses of TQ (20 mg/kg) resulted in a significant reduction in BUN, uric acid, and creatinine compared to the CFZ group (*p* < 0.0001). In comparison to the control group, TQ alone did not provide any statistically significant results (*p* > 0.05). The details are provided in Figure 1A–C.

### 2.2. Impact of TQ Treatment on Malondialdehyde (MDA)

Administration of CFZ (4 mg/kg) resulted in a significant rise in MDA of 167.71% compared to the normal control (*p* < 0.0001). TQ treatment (10 mg/kg) resulted in a significant decrease in MDA of −14.08% compared to the CFZ treatment group (CFZ) (*p* < 0.001). Similarly, the highest doses of TQ (20 mg/kg) resulted in a significant reduction in lipid peroxidation compared to the CFZ group (*p* < 0.0001). In comparison to the control group, TQ alone did not provide any statistically significant results (*p* > 0.05) (Figure 2).

### 2.3. Impact of TQ Treatment on Antioxidants (GSH, CAT, and SOD)

Administration of CFZ (4 mg/kg) resulted in a significant (*p* < 0.0001) reduction in glutathione GSH (−61.51%), catalase CAT (−50.76%), and superoxide dismutase (SOD) (−56.41%) as compared to the normal control.

TQ treatment (10 mg/kg) resulted in a significant rise in GSH (*p* < 0.001), CAT (*p* < 0.0001), and SOD (*p* < 0.0001) compared to the CFZ treatment group. Similarly, the highest doses of TQ (20 mg/kg) resulted in a significant increase in GSH, CAT, and SOD content compared to the CFZ group (*p* < 0.0001). In comparison to the control group, TQ alone did not provide any statistically significant results (*p* > 0.05) (Figure 3A–C).

### 2.4. Effect of TQ Treatment on Gene Expression (IL-1β and IL-6)

IL-1β and IL-6 gene expressions were increased by 4 mg/kg of CFZ compared to the normal control (*p* < 0.001). TQ treatment (10 and 20 mg/kg) reduced IL-1β and IL-6 gene expressions in rat kidneys following CFZ administration (*p* < 0.01, *p* < 0.001). No significant gene expression was seen in treatment with TQ20 alone as compared to the normal control (*p* > 0.05) (Figure 4A,B).

### 2.5. Effect of TQ Treatment on Tumor Necrosis Factor (TNF-α) Production

After administering CFZ (4 mg/kg), the levels of TNF were five times higher than in the normal control group (*p* < 0.0001). Treatment with TQ10 and TQ20 significantly reduced the production of TNF-α dose-dependently in comparison to the corresponding CFZ group (*p* < 0.0001). In addition, the concentration of TNF-α was not significantly altered in the TQ20 compared to the normal control (Figure 5).

### 2.6. Effect of TQ Treatment on an Apoptotic Marker (Caspase-3)

Apoptotic marker (caspase-3) concentrations were increased three-fold above the normal control following CFZ (4 mg/kg) administration (*p* < 0.0001). Pre-administration of TQ dose-dependently reduced the caspase-3 levels following CFZ administration (*p* < 0.0001). On the contrary, caspase-3 concentration was not significantly altered in the TQ-treated group (20 mg/kg) compared to the normal control group (Figure 6).

### 2.7. Effects of Thymoquinone on Nrf2 Gene Expression

We investigated Nrf2 expression in rat kidneys following CFZ administration. We observed that CFZ administration significantly reduced Nrf2 expression (*p* < 0.0001). TQ enhanced Nrf2 expression (*p* < 0.001), and this effect was only seen with 20 mg/kg. Nrf2 expression enhancement with 10 mg/kg did not show a significant level compared to CFZ administration (*p* > 0.05). Moreover, TQ (20 mg/kg) alone significantly enhanced Nrf2 expression compared to the normal control group (*p* < 0.01) (Figure 7).

### 2.8. Effects of Thymoquinone on Histopathological Examination

Histological analysis of the normal and TQ20-treated groups revealed healthy kidney structures devoid of pathological abnormalities, including unaltered glomeruli and tubules (Figure 8A–E). However, the CFZ group had tubular vacuolization and necrosis, as well as degenerative alterations in the glomerular basement membrane (Figure 8B). Degenerative kidney alterations caused by CFZ were reduced with TQ treatment (10 and 20 mg/kg; Figure 8C,D). We looked at the damage to the tubules and assigned the tubular epithelial damage in the renal cortex a score between 0 (normal) and 4 (severe). Tubular cell necrosis, glomerular cell degeneration, vacuolization, tubular cell enlargement, etc., were used as criteria for the scoring system. Score “0” represents no changes; score “1” represents minor changes; score 2 represents mild changes; scores 3–4 represent severe changes.

## 3. Discussion

Carfilzomib (CFZ), a proteasome inhibitor of the second generation, is used to treat multiple myeloma. With the exception of the brain, CFZ is transported to all tissues and rapidly destroyed by enzymatic hydrolysis, resulting in the formation of inactive metabolites [20,21]. Despite its therapeutic efficacy, CFZ has been related to cardiorenal side effects. Nonetheless, the molecular pathways behind CFZ-associated nephrotoxicity are unknown. Preclinical investigations have revealed putative prophylactic medicines for CFZ-induced cardiorenal issues, but none have been used in clinical practice. Determining the mechanisms behind CFZ-induced kidney damage is thus an unmet therapeutic need.

Thymoquinone (TQ) is a major active compound present in the black seeds of *Nigella sativa* (Ranunculaceae) and the aerial parts of Monarda fistulosa (Lamiaceae). It is nontoxic and has promising pharmacological properties against several illnesses. It exhibits outstanding antioxidant, anti-inflammatory, anticancer, and other important biological activities [22,23,24,25]. 

Fotiou et al., 2020, stated that there are two plausible ways to explain how CFZ affects the kidneys: first, damage to the endothelium of the blood vessels caused by CFZ, and second, over-activation of the complement membrane attack complex [7]. As a result, an increase in plasma creatinine and BUN was identified as an indication of renal failure. Similarly, Efentakis et al.’s 2022 investigation found that CFZ treatment at 4 and 8 mg/kg was linked with significant acute renal damage in C57BI/6J mice, indicated biochemically by an increase in blood creatinine and lactate dehydrogenase (LDH) [10]. Moreover, our study revealed that 4 mg/kg of CFZ for 16 days resulted in a significant rise in the levels of plasma creatinine, blood urea nitrogen (BUN), and uric acid compared to the vehicle-treated group in rats. However, treatment with TQ10 and TQ20 significantly improved the alterations in kidney function by dropping these marker levels toward normal in the serum. This might be due to the antioxidant effect contributed by TQ to the nephrons.

Glutathione is a major antioxidant in the body and plays a decisive role in the detoxification of reactive oxygen species (ROS) [26]. GSH depletion is a significant contributor to lipid peroxidation, which is mediated by ROS-induced cellular damage [27,28,29]. Our study showed that CFZ administration for 16 days significantly reduced GSH in rat kidney cells. In addition, the malondialdehyde (MDA) level, a marker of lipid peroxidation level, was also increased following CFZ administration. Furthermore, the levels of other antioxidant enzymes, catalase (CAT) and superoxide dismutase (SOD), decreased significantly after CFZ administration. This may be due to the damaging effects of CFZ on the renal tubules, manifested by increasing the oxidative stress that causes nephrotoxicity. Nonetheless, treatment with TQ dose-dependently lowered lipid peroxidation and greatly raised glutathione (GSH) content in the rat kidneys, both of which were suggestive of a robust antioxidant activity that effectively mitigates CFZ-induced free radical renal damage. TQ also significantly boosted the activity of the antioxidant enzymes, indicating a minimization in lipid peroxidation that can be immediately seen. This result indicates that TQ has the potential to neutralize ROS, minimizing the damage to the lipid membranes of the kidney.

To further investigate the nephroprotective effect of TQ treatment, gene expression and production of inflammatory and apoptotic mediators were measured following CFZ administration. Previous investigations showed that CFZ stimulated the production and expression of inflammatory cytokines via the iNOS-mediated NF-κB signaling pathway in Wistar albino rats [30]. Al Harbi et al.’s 2019 study also showed that CFZ at a dose of 4 mg/Kg activated caspase-3-mediated apoptosis in rats [11]. Similarly, we observed that CFZ administration enhanced the concentration of IL-1β, IL-6, TNF-α, and caspase-3 in tissue and in serum. However, the levels of these signals were downregulated by TQ treatment in rat kidney cells following CFZ administration.

Erythroid-related nuclear factor 2 (Nrf2) regulates the basal expression of many antioxidants’ response element-dependent genes to neutralize the oxidative stress affected by ROS [31]. Recent reports showed that TQ treatment protects against cisplatin-induced kidney injury by ameliorating Nrf2/HO-1 mediated signaling [32]. Interestingly, our study revealed that TQ administration significantly mitigated the negative effects of CFZ treatment on Nrf2 expression, demonstrating the drug’s potential as a nephroprotective prophylactic strategy to mitigate the damage caused by CFZ.

Histopathological changes and the amelioration effect of TQ, along with CFZ administration, also supported the aforementioned biochemical findings. The kidney tissue samples from CFZ-treated rats showed severe degeneration, substantial inflammatory cell filtration, and necrosis in the majority of the glomerular basement membranes. The tissue sections of the group treated with TQ alone, on the other hand, showed no histological alterations. An interesting finding was that the kidney portions of both treated groups remarkably started to recover from the injury.

TQ is known for its high-quality antioxidant and anti-inflammatory activity, which was reflected in this study. Several investigations have shown that polyphenols have antioxidant capabilities, including the capacity to block NADPH oxidase activation, the primary cellular generator of superoxide [33]. Collectively, this study showed that TQ treatment effectively protected against CFZ-associated renal impairment and renal injury in rat kidney cells by altering several key signaling proteins, such as the Nrf2-redox system’s ability to regulate NF-κB-mediated inflammatory cytokines and caspase-3 activation.

## 4. Material and Methods

### 4.1. Reagents and Chemicals

Blood urea nitrogen, uric acid, and creatinine biochemical marker kits were bought from Crescent Diagnostic in Jeddah, Saudi Arabia, 21423. TNF- and Caspase-3 ELISA kits were bought from MyBioSource in the US through a supplier in Saudi Arabia. Genes of interest, including IL-1β, IL-6, Nrf2, and β-actin, were obtained from Macrogen, Korea. Table 1 shows the PCR primers that were used in this study. The following ingredients were purchased from Sigma Aldrich Co., St. Louis, MO, USA: thymoquinone, carfilzomib, thiobarbituric reactive material, reduced glutathione, 5,5-dithiobis-(2-nitrobenzoic acid), sulfosalicylic acid, trichloroacetic acid, and hydrogen peroxide.

### 4.2. Experimental Scheme

For this investigation, male Wistar albino rats weighing between 150 and 180 g were purchased from the Jazan University Medical Research Centre animal house in Jazan, Saudi Arabia. Throughout the trial, the rats were kept in optimal lab settings, receiving a daily pellet meal and having free access to water. The animals used in this study were all healthy and had not undergone any cancer therapies or been diagnosed with multiple myeloma. The Institutional Research Review and Ethics Committee approved an experimental investigation with the reference number REC42/1/128.

Randomly, 30 male Wistar albino rats were allocated into five groups containing six rats each. The CFZ drug was administered via intraperitoneal injection, while TQ was administered by oral gavage. The dosing schemes of different groupings are presented in Figure 9 below.

At the end of the experiment (day 17), we collected blood samples. After administering anesthesia, blood was extracted from the animals via orbital puncture and collected in tubes. After holding the blood at 37 °C for 15 to 30 min, the blood clot was removed by centrifuging it at 2000 rpm for 10 min in a refrigerated centrifuge, and serum was collected. Further serum was stored at 4 °C for hematological assay (creatinine, blood urea nitrogen, and uric acid), while kidney tissue was isolated for further histopathological analysis and tissue homogenization for the collection of supernatants to analyze the oxidative stress markers (LPO. GSH, CAT, and SOD) and inflammatory cytokine markers (TNFα, Caspase-3, IL-1β, IL-6, and Nrf2) assay.

### 4.3. Sample Preparation (Kidney Tissue)

Sample preparation of kidney tissue was carried out as per Alam et al.’s 2022 procedure [34]. Briefly, to test oxidative stress (GSH, CAT, SOD, MDA, etc.), a 10% homogenate of kidney tissue was prepared by homogenizing the tissue in a phosphate-buffered solution of 0.1 M and pH 7.4. The PMS was then obtained by centrifugation. The kidney homogenate was made with 1 g/mL of protease inhibitor in phosphate buffer (pH 7.4). After that, the material was centrifuged at 4 °C for 5 min at 800× *g*. The homogenate used for the LPO and GSH analyses was separated from the supernatant (S1). The leftover homogenate was centrifuged at 10,500× *g* for 15 min at 4 °C to separate out the post-mitochondrial supernatant (PMS). After that, we used the PMS to test for SOD and CAT. 

### 4.4. Kidney Function Marker Assay

Serum levels of creatinine, blood urea nitrogen, and uric acid were analyzed using Crescent diagnostic testing kits to assess kidney function. Absorbance measurements were taken of both the test specimen and the standard, and the values of each parameter were determined using the formulas provided in the kit.

### 4.5. Antioxidant Assay (Lipid Peroxidation, Glutathione, Catalase, and Superoxide Dismutase)

Using the method indicated by Lowry et al. (1951), the protein content of each sample was determined [35]. The LPO concentration was measured using the technique described by Islam et al., 2002. In short, 0.5 mL of homogenate were kept at 37 °C for 1 h in a metabolic shaker, and an equal amount of homogenate was kept at 0 °C for the same amount of time. After incubation, 0.5 mL of cold 5% (*w*/*v*) TCA and 0.5 mL of 0.67% TBA were added, and the mixture was spun for 10 min at 4000× *g*. Next, the supernatant was placed into test tubes and in a bath of hot water for 10 min. After that, a pink color was created, and a spectrophotometer (Shimadzu-1601, Kyoto, Japan) was used to test it at 535 nm. The amount of TBARS was determined using a molar extinction coefficient of 1.56 × 10^5^ M^−1^ cm^−1^ and expressing it as the number of nanomoles of TBARS made per hour per gram of protein [36].

The glutathione assay was performed in accordance with Jollow et al., 1974. In addition, the sample was combined with 4% (*w*/*v*) sulfosalicylic acid in a 1:1 (*v*/*v*) ratio. After 1 h of incubation at 4 °C, the samples were centrifuged at 4000*g* for 10 min at 4 °C. In addition, 1.0 mM DTNB was added to the assay mixture and the resulting yellow color was measured at 412 nm. Using a molar extinction coefficient of 13.6 × 10^3^ M^−1^ cm^−1^, the GSH concentration was computed as micromoles of GSH mg-1 protein [37].

Catalase (CAT) activity was determined using Claiborne’s methods [38]. At 240 nm, the change in absorbance was measured. Using the molar extinction coefficient of 43.6 × 10^3^ M^−1^ cm^−1^, catalase activity was determined in units of nanomole H_2_O_2_ consumed/min/mg protein.

Superoxide dismutase (SOD) activity was obtained using Marklund’s methods at 580 nm wavelength [39]. The enzyme activity is expressed in units per mg of protein, where 1 unit of enzyme activity corresponds to a 50% reduction in pyrogallol autoxidation.

### 4.6. RT-PCR Assay to Isolate RNA and to Detect Gene Expression of Nrf2, IL-1β, and IL-6

Total RNA was extracted from rat kidneys using TRIzol reagent (Thermo Fisher Scientific, Waltham, MA, USA), and RNA quantities were determined using a NanoDrop ND-2000 spectrophotometer (NanoDrop Technologies, Wilmington, NC, USA). The cDNA was synthesized using the Universal RT-PCR Kit (Solarbio, Beijing, China).

Quantitative PCR using SYBR Green was used to determine gene expression (qPCR). Quantitative polymerase chain reaction (qPCR) using a 2X SYBR Green PCR Matrix was performed on a 7300 Applied Biosystems Step One PlusReal-Time PCR System (Thermo Fisher Scientific) (Solarbio, China). The PCR cycling conditions were as follows: initial denaturation at 95 °C for 2 min, followed by 30 cycles of denaturation at 95 °C for 15 s, annealing at 60 °C for 1 min, and extension at 72 °C for 15 s. To normalize expression, the housekeeping gene β-actin was employed as a control. The relative quantification (RQ) formula was used to calculate gene expression levels. The target gene cycle threshold (Ct) value was normalized to β-actin’s Ct value in the same sample, and the relative expression was calculated using the 2^−ΔΔCt^ method [40]. 

### 4.7. ELISA Assay for Tumor Necrosis Factor (TNF-α)

The Rat TNF-α ELISA Kit (ab236712) is a single-wash, 90 min sandwich ELISA used to quantify TNF-α protein in serum according to the manufacturer’s instructions. Using an ELISA microplate reader (BioTek ELx800, Saint Charles, IL, USA), the absorbance of interleukin TNF-α was measured at 450 nm. In brief, samples or standards are placed into the wells first, followed by the antibody mix. The wells were cleansed after incubation to eliminate loose material. A further development solution was added to catalyze the reaction during incubation, resulting in a blue color. The reaction was subsequently terminated by the addition of a stop solution, which converted the blue color to yellow. The intensity of the signal is measured at 450 nm and is proportional to the amount of bound analyte.

### 4.8. ELISA Assay for Apoptotic Marker (Caspase-3)

ELISA (enzyme-linked immunosorbent assay) kits for detecting caspase 3 were used to target the analytes in biological samples as per the standard protocol of the kits. The ELISA analytical biochemical approach was used to detect caspase-3 antigen targets in samples based on caspase-3 antibody-caspase-3 antigen contacts (immunosorbency) and an HRP colorimetric detection system at 405 nm.

### 4.9. Histopathological Examination

Immediately after the animals were slaughtered, samples of kidney tissue were removed and cleaned with ice-cold normal saline that had a 0.9% concentration. After that, a solution containing 10% formalin was used to fix the tissues. Additional tissue was implanted into liquid paraffin and then shaped into slabs. Then, a microtome was used to slice these blocks into segments spanning in thickness from 3 to 5 μm. The prepared samples were stained with haematoxylin and eosin (H&E) and observed under a 40× magnification microscope. This was conducted in order to continue the histological analysis.

### 4.10. Statistical Analysis

An analysis of variance was used to statistically examine the data, and then the Tukey–Kramer test was applied. The data were presented using the mean standard error of six rats (SEM). The data were evaluated for statistical significance using the *p* < 0.05 threshold.

## 5. Conclusions

The findings of this study indicated that CFZ-induced renal impairment is considerably influenced by the antioxidant system, lipid peroxidation, activation of inflammatory cytokines, caspase-3, and downregulation of gene expression (Nrf2). Thymoquinone (TQ) was shown to protect the kidneys against CFZ-induced renal impairment by lowering kidney injury markers, boosting the antioxidant system, and preventing the production and expression of pro-inflammatory genes. Furthermore, TQ administration significantly mitigated the negative effects of CFZ treatment on the antioxidant defense system. Thus, to reduce the negative effects of CFZ, TQ may provide a unique and targeted renoprotective agent for chemotherapeutic medications.

## Figures and Tables

**Figure 1 ijms-24-10621-f001:**
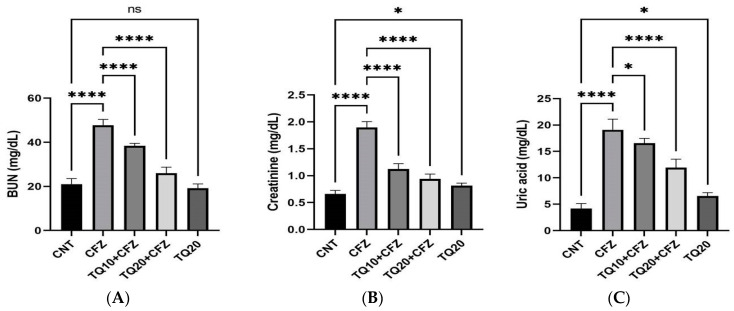
Impact of thymoquinone treatment on kidney function markers. Data were displayed as mean ± standard error mean (*n* = 6) along with significant value **** *p* < 0.0001 (CFZ vs. CNT; TQ10 + CFZ vs. CFZ; TQ20 + CFZ vs. CFZ) and ^ns^ *p* > 0.05 (TQ20 vs. CNT) in (**A**,**B**). The significant value in (**C**) was **** *p* < 0.0001(CFZ vs. CNT; TQ20 + CFZ vs. CFZ) and * *p* < 0.01 (TQ10 + CFZ vs. CFZ and TQ20 vs. CNT). Abbreviations—CNT: control, CFZ: carfilzomib, and TQ: thymoquinone.

**Figure 2 ijms-24-10621-f002:**
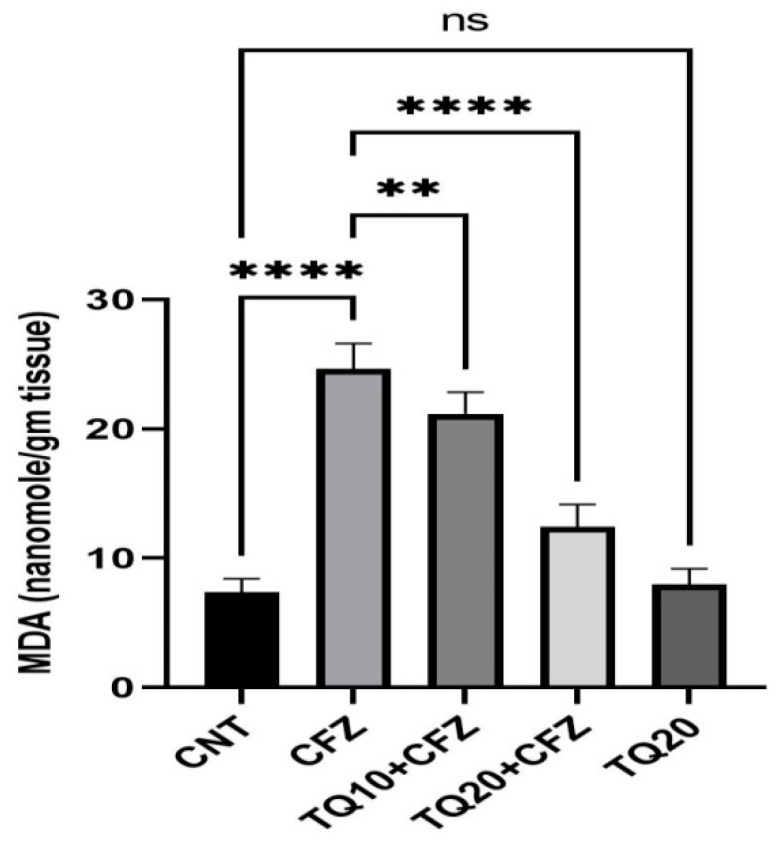
Impact of thymoquinone treatment on lipid peroxidation (MDA). Data were displayed as mean ± standard error mean (*n* = 6) along with significant value **** *p* < 0.0001 (CFZ vs. CNT; TQ20 + CFZ vs. CFZ), ** *p* < 0.001 (TQ10 + CFZ vs. CFZ), and ^ns^ *p* > 0.05 (TQ20 vs. CNT). Abbreviations—CNT: control, CFZ: carfilzomib, and TQ: thymoquinone.

**Figure 3 ijms-24-10621-f003:**
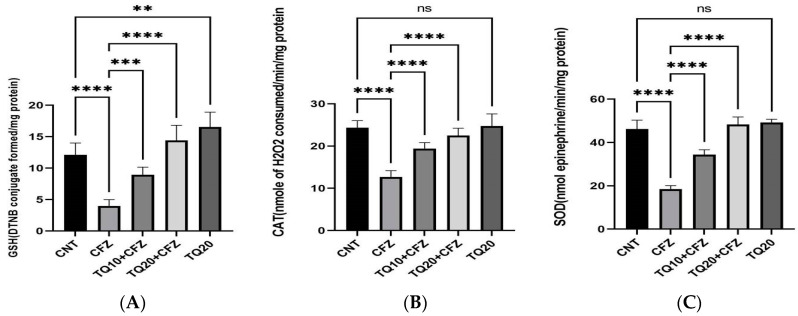
Impact of thymoquinone treatment on antioxidant enzymes. Data were displayed as mean ± standard error mean (*n* = 6) along with significant value **** *p* < 0.0001 (CFZ vs. CNT) (TQ20 + CFZ vs. CFZ), *** *p* < 0.001 (TQ10 + CFZ vs. CFZ), ** *p* < 0.001 (TQ20 vs. CNT) in (**A**) The significant value in (**B**,**C**) was **** *p* < 0.0001 (CFZ vs. CNT), (TQ10 + CFZ vs. CFZ), (TQ20 + CFZ vs. CFZ); ^ns^ *p* > 0.05 (TQ20 vs. CNT). Abbreviations—CNT: control, CFZ: carfilzomib, and TQ: thymoquinone.

**Figure 4 ijms-24-10621-f004:**
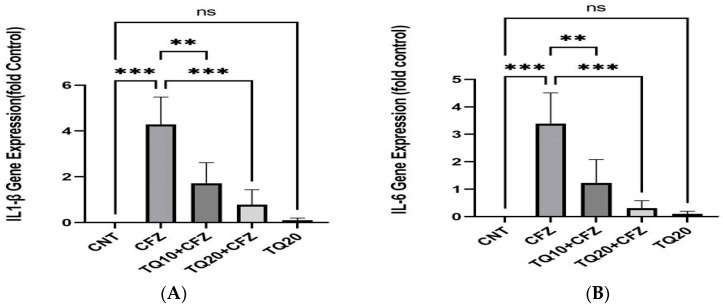
Impact of thymoquinone treatment on IL-1β and IL-6 gene expression. The cycle threshold (Ct) value of target genes was normalized to the Ct value of β-actin in the same sample. Data were displayed as mean ± standard error mean (*n* = 6) along with significant value *** *p* < 0.001 (CFZ vs. CNT; TQ20 + CFZ vs. CFZ), ** *p* < 0.001 (TQ10 + CFZ vs. CFZ), and ^ns^ *p* > 0.05 (TQ20 vs. CNT) in (**A**) and (**B**). Abbreviations—CNT: control, CFZ: carfilzomib, and TQ: thymoquinone.

**Figure 5 ijms-24-10621-f005:**
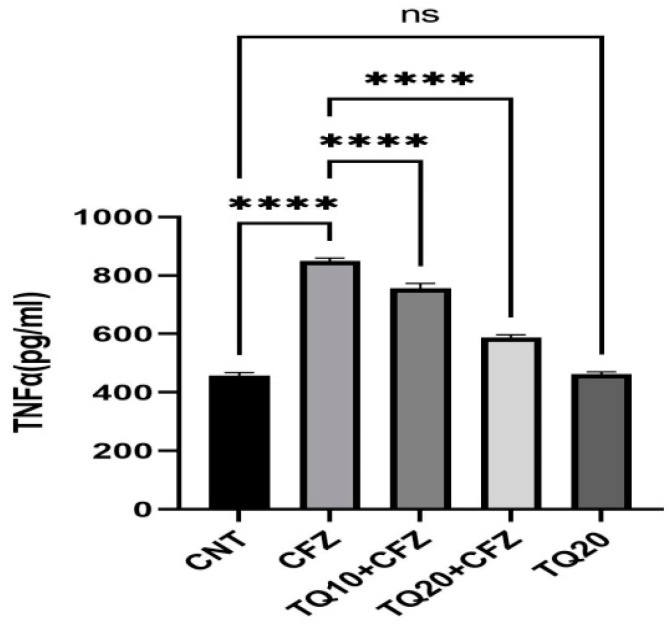
Impact of thymoquinone treatment on tumor necrosis factor (TNF-α). Data were displayed as mean ± standard error mean (*n* = 6) along with significant value **** *p* < 0.0001 (CFZ vs. CNT; TQ10 + CFZ vs. CFZ; TQ20 + CFZ vs. CFZ) and ^ns^ *p* > 0.05 (TQ20 vs. CNT). Abbreviations—CNT: control, CFZ: carfilzomib, and TQ: thymoquinone.

**Figure 6 ijms-24-10621-f006:**
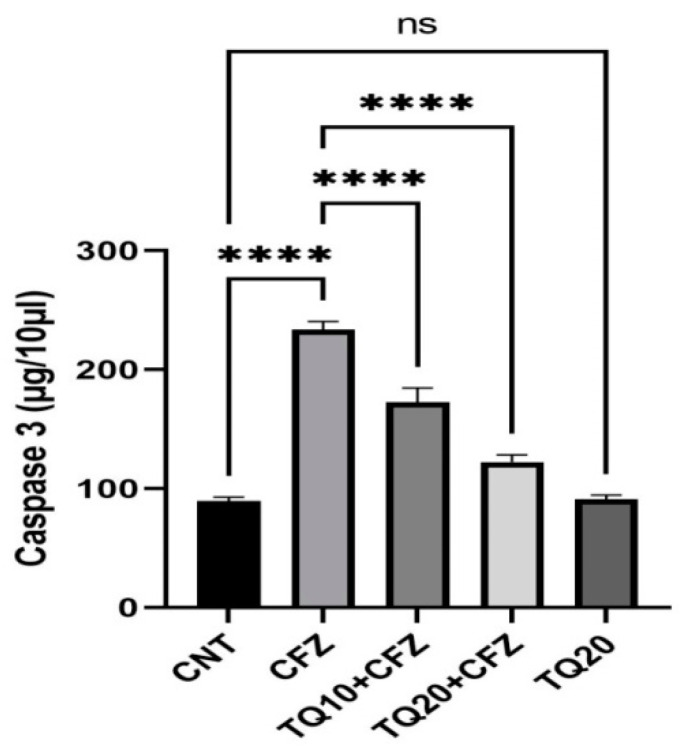
Impact of thymoquinone treatment on apoptotic marker (caspase3). Data were displayed as mean ± standard error mean (*n* = 6) along with significant value **** *p* < 0.0001 (CFZ vs. CNT; TQ10 + CFZ vs. CFZ; TQ20 + CFZ vs. CFZ) and ^ns^ *p* > 0.05 (TQ20 vs. CNT). Abbreviations—CNT: control, CFZ: carfilzomib, and TQ: thymoquinone.

**Figure 7 ijms-24-10621-f007:**
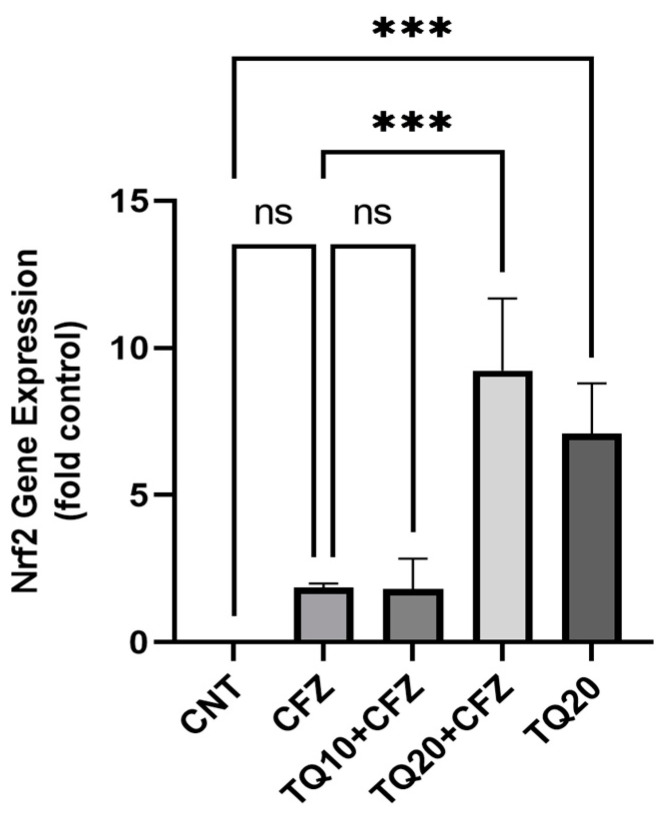
Impact of thymoquinone treatment on Nrf2 gene expression following CFZ treatment. The cycle threshold (Ct) value of the Nrf2 gene was normalized to the Ct value of β-actin in the same sample. Data were displayed as mean ± standard error mean (*n* = 6) along with significant value; *** *p* < 0.001 (TQ20 + CFZ vs. CFZ; TQ20 vs. CNT) and ^ns^ *p* > 0.05 (CFZ vs. CNT; TQ10 + CFZ vs. CFZ). Data represent mean ± S.E M. of 6 animals per treatment group. Abbreviations—CNT: control, CFZ: carfilzomib, and TQ: thymoquinone.

**Figure 8 ijms-24-10621-f008:**
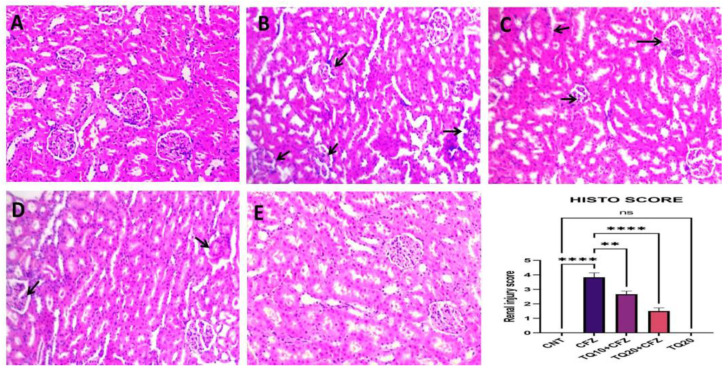
Histological analysis of the normal control indicated without any abnormalities in the glomeruli basement membrane and renal tubules with zero score (**A**). TQ-treated group revealed healthy kidney structure devoid of pathological abnormalities, including unaltered glomeruli basement membrane and renal tubules with zero score (**E**). However, the CFZ group indicated arrow tubular vacuolization, necrosis, as well as degenerative alterations in the glomerular basement membrane with renal injury score = 4 (**B**). Degenerative kidney alterations caused by CFZ were reduced with TQ treatment (10 and 20 mg/kg) with renal injury score = 2 and score = 1 for (**C**) and (**D**), respectively. ** *p* < 0.01 (TQ10 + CFZ vs. CFZ), **** *p* < 0.0001 (TQ20 + CFZ vs. CFZ), and ^ns^
*p* > 0.05 (TQ20 vs. CNT). Arrow (
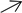
) indicates the glomerular basement membrane degeneration, tubular vacuolization and necrosis, etc.

**Figure 9 ijms-24-10621-f009:**
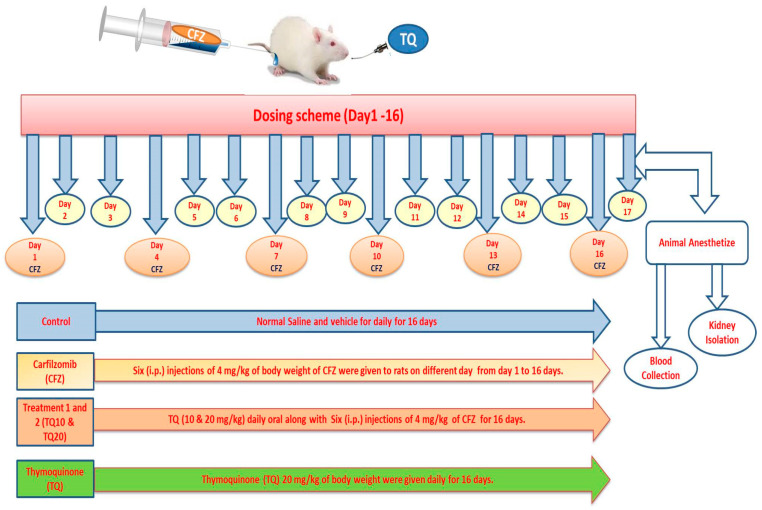
Male Wistar albino rats grouping and dosing scheme.

**Table 1 ijms-24-10621-t001:** Sequences of PCR primers.

Primer Name	Forward (5′ to 3′)	Reverse (5′ to 3′)
Rat β-Actin	CTTGCAGCTCCTCCGTCGCC	CTTGCTCTGGGCCTCGTCGC
Rat IL-1β	CTGTGACTCGTGGGATGATG	AGGGATTTTGTCGTTGCTTG
Rat IL-6	AGTTGCCTTCTTGGGACTGA	ACAGTGCATCATCGCTGTTC
Rat Nrf2	CTCTCTGGAGACGGCCATGACT	CTGGGCTGGGGACAGTGGTAGT

## Data Availability

The authors affirm that the data are included in the manuscript for publication.

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
