# Peer review of "Thymoquinone Ameliorates Carfilzomib-Induced Renal Impairment by Modulating Oxidative Stress Markers, Inflammatory/Apoptotic Mediators, and Augmenting Nrf2 in Rats"

_ijms, 2023, doi:10.3390/ijms241310621_

Round 1

Reviewer 1 Report (New Reviewer)

the manuscript is interesting but it presents some flaws that must be resolved. In particular:

Line 52: CFZ?

Introduction: since NRF2 plays a key role in this manuscript, an introduction about its role and function deserves to be added. In fact, this trascription factor plays a key role in preventing cancer onset and progression (see  PMID: 3664110036335520,   35901941, 36289931 )

Figure 8: Figure quality is very low and must be improved. Magnifications of interested points must be added. 

Line 310: 4 degrees Celsius= 4° C

An accurate revision of typing errors and syntax is recommended

Figure numbers must be removed from the figure since it is already written in the legend

Acronyms must be written in full length when mentioned for the first time

An accurate revision of typing errors and syntax is recommended

Author Response

Response to Reviewer-1 Comments                 date 13/06/2023

Open Review

Dear Sir,

We sincerely appreciate your input and thank you for giving us the opportunity to improve the clarity and quality of our study. As per your suggestion I have modified the manuscript and hope you will found it upto the mark.

Quality of English Language

( ) I am not qualified to assess the quality of English in this paper
( ) English very difficult to understand/incomprehensible
( ) Extensive editing of English language required
( ) Moderate editing of English language required
(x) Minor editing of English language required
( ) English language fine. No issues detected

Yes

Can be improved

Must be improved

Not applicable

Does the introduction provide sufficient background and include all relevant references?

( )

( )

(x)

( )

Are all the cited references relevant to the research?

( )

(x)

( )

( )

Is the research design appropriate?

( )

(x)

( )

( )

Are the methods adequately described?

( )

(x)

( )

( )

Are the results clearly presented?

( )

( )

(x)

( )

Are the conclusions supported by the results?

( )

(x)

( )

( )

Comments and Suggestions for Authors

the manuscript is interesting but it presents some flaws that must be resolved. In particular:

Line 52: CFZ?

Response: Carfilzomib

Introduction: since NRF2 plays a key role in this manuscript, an introduction about its role and function deserves to be added. In fact, this trascription factor plays a key role in preventing cancer onset and progression (see  PMID: 36641100 , 36335520,  35901941, 36289931 )

Response: Added in introduction parts and cited of the following articles.

Tossetta G, Marzioni D. Targeting the NRF2/KEAP1 pathway in cervical and endometrial cancers. Eur J Pharmacol. 2023 Feb 15;941:175503. doi: 10.1016/j.ejphar.2023.175503. Epub 2023 Jan 12. PMID: 36641100.

Ghareghomi S, Habibi-Rezaei M, Arese M, Saso L, Moosavi-Movahedi AA. Nrf2 Modulation in Breast Cancer. Biomedicines. 2022 Oct 21;10(10):2668. doi: 10.3390/biomedicines10102668. PMID: 36289931; PMCID: PMC9599257.

Figure 8: Figure quality is very low and must be improved. Magnifications of interested points must be added. 

Response: As reviewer comments Figure quality were improved and 300DPI

Line 310: 4 degrees Celsius= 4° C

Response: Corrected

An accurate revision of typing errors and syntax is recommended

Response: Corrected typing errors and syntax

Figure numbers must be removed from the figure since it is already written in the legend

Response: Deleted the figure number from the legend

Acronyms must be written in full length when mentioned for the first time

Response:Corrected

Comments on the Quality of English Language

An accurate revision of typing errors and syntax is recommended

Response: Typo error checked and corrected

Submission Date

28 May 2023

Date of this review

07 Jun 2023 11:27:43

Reviewer 2 Report (New Reviewer)

The present manuscript contains several aspects of improvement, ranging from the methodology to the interpretation of the results. In the following, I describe those that I consider to be the most important:

Abstract

Very imprecise, they do not explain for how long what they say is pre-treatment is administered (I will talk about this later), they generalize antioxidants with GSH

Introduction

between lines 46 and 47 a part of the sentence is repeated “complication that decreases the efficacy of 46 cancer treatment”

Line 49: acute kidney injury manifests itself through the increase of certain markers in the blood, it does not manifest itself as acute kidney injury.

Line 51 it would be interesting if at this point the incidence or prevalence of aki in patients receiving this treatment is discussed

Results

Why are there no baseline data on kidney damage biomarkers to ensure that the animals start from a healthy baseline condition?

Are there no significant differences between TQ10+CFZ and CNT in bun and uric acid? With an n of 5 and such a low error, my observation is that this difference does exist. Have the statistical tests been applied correctly? This same appreciation could be made for practically all the markers used

How is the very different profile of Nrf2 gene expression compared to the other markers explained? What does that elevation mean in TQ20? Later in the discussion this is not explained.

In Figure 8, explain in the caption which group corresponds to each letter. what do the arrows point to? There should be a different type of arrow or marker for each type of damage to help to understand the type of damage associated with each treatment.

If you try to explain that there is less inflammation, why hasn't immunohistochemistry been performed to see which areas improve the administration of the substance under test or if there is a dose-dependent effect as it appears with the biomarkers analyzed?

Albuminuria is evaluated in patients and is associated with glomerulosclerosis in the renal biopsy. Why has it not been measured in this study?

This study is done in animals without cancer, it would be interesting to test it in animals with myeloma and not only evaluate the effect on the kidney but also whether or not the administration of this substance modifies the anticancer activity

Discussion

Much information is repetition from the introduction.

The importance of understanding the associated mechanisms of CFZ-induced renal toxicity is discussed, but in this paper they focus on a mechanism that has already been demonstrated.

“pretreatment with TQ significantly improved the alterations in kidney function by dropping these marker 226 levels toward normal in the serum” (líneas 225-226) But this is not true, in no case do they return to the control values (it is not possible to speak of basal values as they are not measured) and they even remain well above the values considered normal, at least at a dose of 10 mg.

Material and methods

The experimental scheme is not understood. Are six doses given for 16 days? Why is there talk of pretreatment with TQ if it is given at the same time as the toxicant? In this case it would be co-treatment

be careful, in the figure the days 11 and 14 (among others) the numbers have been separated and it seems that it is day 1 with a subtime of 1 or 4

How was the oral administration done? I understand from the photo that by means of a gastric tube, but it is not clear from the text. If the probe is used for so many days, it is necessary to measure infection markers since this pathway can cause this symptom and affect the results, especially when oxidative stress and inflammation markers are being measured.

How is the serum sample obtained? Because according to the protocol it seems that serum is obtained directly from the animals, but this is not correct. Does blood freeze as such? clotted?

Why is the cytokines analysis done in blood and not in kidney or in both samples? The elevation of cytokines in the blood may be due to other non-renal causes, such as inflammation produced by the cannula itself.

4.4 kidney function marker assay I understand that this is done in serum but it must be specified in this section. The same or for point 4.5 explain the sample, although it has already been mentioned previously, it facilitates the interpretation for the reader

Line 383 in histology appears "After some time had passed" very imprecise, the time is days, hours, weeks?

English can be improved, there are colloquial expressions. Some sentences are difficult to understand.

Author Response

Response to Reviewer-2 Comments    Date 13/06/2023

Dear Sir,

We sincerely appreciate your input and thank you for giving us the opportunity to improve the clarity and quality of our study. As per your suggestion I have modified the manuscript and hope you will found it up to the mark.

Open Review

Quality of English Language

( ) I am not qualified to assess the quality of English in this paper
( ) English very difficult to understand/incomprehensible
( ) Extensive editing of English language required
(x) Moderate editing of English language required
( ) Minor editing of English language required
( ) English language fine. No issues detected

Yes

Can be improved

Must be improved

Not applicable

Does the introduction provide sufficient background and include all relevant references?

( )

(x)

( )

( )

Are all the cited references relevant to the research?

(x)

( )

( )

( )

Is the research design appropriate?

( )

(x)

( )

( )

Are the methods adequately described?

( )

(x)

( )

( )

Are the results clearly presented?

( )

(x)

( )

( )

Are the conclusions supported by the results?

( )

(x)

( )

( )

Comments and Suggestions for Authors

The present manuscript contains several aspects of improvement, ranging from the methodology to the interpretation of the results. In the following, I describe those that I consider to be the most important:

Abstract

Very imprecise, they do not explain for how long what they say is pre-treatment is administered (I will talk about this later), they generalize antioxidants with GSH

Introduction

between lines 46 and 47 a part of the sentence is repeated “complication that decreases the efficacy of 46 cancer treatment”

Response: Deleted the repeated sentence

Line 49: acute kidney injury manifests itself through the increase of certain markers in the blood, it does not manifest itself as acute kidney injury.

Response: Corrected

Line 51 it would be interesting if at this point the incidence or prevalence of aki in patients receiving this treatment is discussed.

Response: I disagree with this suggestion because we are highlighting the issue rather than treatment because it is part of a Introduction where we highlighted the issue.

Results

Why are there no baseline data on kidney damage biomarkers to ensure that the animals start from a healthy baseline condition?

Response: There is already base line data available for all kidney damage biomarkers in the healthy and normal animals and compared with Carfilzomib associated kidney damage.

Are there no significant differences between TQ10+CFZ and CNT in bun and uric acid? With an n of 5 and such a low error, my observation is that this difference does exist. Have the statistical tests been applied correctly? This same appreciation could be made for practically all the markers used.

Response: I did not compare the results of BUN and Uric acid between TQ10+CFZ and CNT. I compared the results of BUN and Uric acid in between TQ10+CFZ and CFZ. I did not use the n=5 for statistical analysis. I used n=6  and mentioned in the legend of the figure.

How is the very different profile of Nrf2 gene expression compared to the other markers explained? What does that elevation mean in TQ20? Later in the discussion this is not explained.

In Figure 8, explain in the caption which group corresponds to each letter. what do the arrows point to? There should be a different type of arrow or marker for each type of damage to help to understand the type of damage associated with each treatment.

Response: Thanks for valuable comments. I have corrected the figure and removed the arrow from normal control and drug TQ20 alone because there was no abnormality was observed. While other three groups arrow indicates the abnormality in glomeruli basement membrane, vacuolization, renal duct damages which was reflected in three groups.

If you try to explain that there is less inflammation, why hasn't  immunohistochemistry been performed to see which areas improve the administration of the substance under test or if there is a dose-dependent effect as it appears with the biomarkers analyzed?

Response:  Thanks for remarkable comments and I was unable to do some of the important test due to lack of financial support. This is supported by the project along with minimum budgets and was not possible to run all the experiments but in future we will do for second project.

Albuminuria is evaluated in patients and is associated with glomerulosclerosis in the renal biopsy. Why has it not been measured in this study?

Response:  Thanks for remarkable comments and I was unable to do some of the important test due to lack of financial support.

This study is done in animals without cancer, it would be interesting to test it in animals with myeloma and not only evaluate the effect on the kidney but also whether or not the administration of this substance modifies the anticancer activity

Response:  I really appreciate your comments and this is our further future plan to evaluate in animals with myeloma to see the double action of TQ against the chemotherapeutic drugs as well as against cancer treatment.

Discussion

Much information is repetition from the introduction.

Response: Those sentences were in repetition further modified.

The importance of understanding the associated mechanisms of CFZ-induced renal toxicity is discussed, but in this paper they focus on a mechanism that has already been demonstrated.

Responses: Based on our findings and mechanistic approaches supported the previous study.

“pretreatment with TQ significantly improved the alterations in kidney function by dropping these marker 226 levels toward normal in the serum” (líneas 225-226) But this is not true, in no case do they return to the control values (it is not possible to speak of basal values as they are not measured) and they even remain well above the values considered normal, at least at a dose of 10 mg.

Responses: It is correct observation that TQ improved the alteration in kidney function by dropping these markers levels towards normal. It does not mean that TQ treatment 100% make the normal. It never seen but it helps to normalize it as compared to toxic. Both doses of TQ (TQ10 and TQ20 mg) are effective but TQ20 was more effective than TQ10. Its happen due to its antioxidant effects.

Material and methods

The experimental scheme is not understood. Are six doses given for 16 days? Why is there talk of pretreatment with TQ if it is given at the same time as the toxicant? In this case it would be co-treatment

Response: Thanks for your valuable comments. No six doses of CFZ was given in between day one to day 16. Yes you are correct and it was not a pretreatment case it was co-treatment only.

be careful, in the figure the days 11 and 14 (among others) the numbers have been separated and it seems that it is day 1 with a subtime of 1 or 4

Response: I have again corrected the figure and separated the number.

How was the oral administration done? I understand from the photo that by means of a gastric tube, but it is not clear from the text. If the probe is used for so many days, it is necessary to measure infection markers since this pathway can cause this symptom and affect the results, especially when oxidative stress and inflammation markers are being measured.

Response: The oral administration was done by oral gavage. There was no chance to develop any stress during animal handling because we are well trained in dosing. We take care of rats during the dosing to avoid any stress on animals.

How is the serum sample obtained? Because according to the protocol it seems that serum is obtained directly from the animals, but this is not correct. Does blood freeze as such? clotted?

Response: After administering anesthesia, blood was extracted from animals via orbital puncture and collected in tubes. After holding the blood at 370C for 15 to 30 minutes, the blood clot was removed by centrifuging it at 2000rpm for 10 minutes in a refrigerated centrifuge and serum collected. Further serum was stored at 40C for biochemical analysis.

Why is the cytokines analysis done in blood and not in kidney or in both samples? The elevation of cytokines in the blood may be due to other non-renal causes, such as inflammation produced by the cannula itself.

Response: Yes, we can do but based on small funding and available resources in the department design the project to execute it on time.

4.4 kidney function marker assay I understand that this is done in serum but it must be specified in this section. The same or for point 4.5 explain the sample, although it has already been mentioned previously, it facilitates the interpretation for the reader

Response: Yes I agree and I have modified in the text accordingly and highlighted.

Line 383 in histology appears "After some time had passed" very imprecise, the time is days, hours, weeks?

 Response: I have modified in the text.

Comments on the Quality of English Language

English can be improved, there are colloquial expressions. Some sentences are difficult to understand.

Response: I agree with reviewer comments and where I feel that sentences are complex try to modify it.

Submission Date

28 May 2023 

Date of this review

 05 Jun 2023 14:12:34

Round 2

Reviewer 1 Report (New Reviewer)

Introduction: since NRF2 plays a key role in this manuscript, an introduction about its role and function deserves to be added. In fact, this trascription factor plays a key role in preventing cancer onset and progression (see  PMID: 36641100 , 36335520,  35901941, 36289931 )

Response: Added in introduction parts and cited of the following articles.

Tossetta G, Marzioni D. Targeting the NRF2/KEAP1 pathway in cervical and endometrial cancers. Eur J Pharmacol. 2023 Feb 15;941:175503. doi: 10.1016/j.ejphar.2023.175503. Epub 2023 Jan 12. PMID: 36641100.

Ghareghomi S, Habibi-Rezaei M, Arese M, Saso L, Moosavi-Movahedi AA. Nrf2 Modulation in Breast Cancer. Biomedicines. 2022 Oct 21;10(10):2668. doi: 10.3390/biomedicines10102668. PMID: 36289931; PMCID: PMC9599257.

The studies (Tossetta et al. and Ghareghomi et al.) mentioned by the authors are not present in the reference list. In fact, references 12 and 13 talk about other subjects that do not interest NRF2. 

Figure 8: Figure quality is very low and must be improved. Magnifications of interested points must be added. 

Response: As reviewer comments Figure quality were improved and 300DPI

Although figure quality has been improved, magnifications of interested points (showed by the arrows) have not been added

Author Response

Response to Reviewer-1 Round-2

Dear Sir,

I'm grateful to you for your helpful advice, and I did try to solve all the problems at first, but I missed some, and you brought them to my attention.  Now that I've read your latest notes, I've fixed and changed things based on what you suggested. I also replaced the references which was not suitable for the contents. I hope you can now find the article in good shape.

Open Review

Quality of English Language

( ) I am not qualified to assess the quality of English in this paper
( ) English very difficult to understand/incomprehensible
( ) Extensive editing of English language required
( ) Moderate editing of English language required
( ) Minor editing of English language required
(x) English language fine. No issues detected

Yes

Can be improved

Must be improved

Not applicable

Does the introduction provide sufficient background and include all relevant references?

( )

(x)

( )

( )

Are all the cited references relevant to the research?

( )

(x)

( )

( )

Is the research design appropriate?

( )

(x)

( )

( )

Are the methods adequately described?

( )

(x)

( )

( )

Are the results clearly presented?

( )

(x)

( )

( )

Are the conclusions supported by the results?

( )

(x)

( )

( )

Comments and Suggestions for Authors

Introduction: since NRF2 plays a key role in this manuscript, an introduction about its role and function deserves to be added. In fact, this trascription factor plays a key role in preventing cancer onset and progression (see  PMID: 36641100 , 36335520,  35901941, 36289931 )

Response: Added in introduction parts and cited of the following articles.

Tossetta G, Marzioni D. Targeting the NRF2/KEAP1 pathway in cervical and endometrial cancers. Eur J Pharmacol. 2023 Feb 15;941:175503. doi: 10.1016/j.ejphar.2023.175503. Epub 2023 Jan 12. PMID: 36641100.

Ghareghomi S, Habibi-Rezaei M, Arese M, Saso L, Moosavi-Movahedi AA. Nrf2 Modulation in Breast Cancer. Biomedicines. 2022 Oct 21;10(10):2668. doi: 10.3390/biomedicines10102668. PMID: 36289931; PMCID: PMC9599257.

The studies (Tossetta et al. and Ghareghomi et al.) mentioned by the authors are not present in the reference list. In fact, references 12 and 13 talk about other subjects that do not interest NRF2.

Response: Thanks for suggestion. I have replaced the previous  11,12 references with new references which talk about Nrf2.

Figure 8: Figure quality is very low and must be improved. Magnifications of interested points must be added. 

Response: As reviewer comments Figure quality were improved and 300DPI

Although figure quality has been improved, magnifications of interested points (showed by the arrows) have not been added.

Reponses: I am agree with you. Now I have added the arrow justification in the figure 8 legends.

Submission Date

28 May 2023

Date of this review

13 Jun 2023 10:12:18

Reviewer 2 Report (New Reviewer)

Although the authors have made an effort to answer almost all the questions raised, the methodological errors have not been corrected, nor is the explanation given sufficient to understand why certain essential determinations have not been made to reach these conclusions. I am referring in particular to the fact that oxidative stress was measured in the blood and not in the supposedly affected tissue. In addition, I am surprised that the ethics committees accepted the collection of blood from the orbital sinus, a practice that is no longer used, and that this is not explained if this sample is obtained at the endpoint and there are other more appropriate methods.

Author Response

Open Review

Response to reviewer-2 Comments round -2

Dear Sir,

I'm grateful to you for your helpful advice, and I did try to solve all the problems at first, but I missed some, and you brought them to my attention.  Now that I've read your latest notes, I've fixed and changed things based on what you suggested. I hope you can now find the article in good shape.

Quality of English Language

( ) I am not qualified to assess the quality of English in this paper
( ) English very difficult to understand/incomprehensible
( ) Extensive editing of English language required
( ) Moderate editing of English language required
( ) Minor editing of English language required
(x) English language fine. No issues detected

Yes

Can be improved

Must be improved

Not applicable

Does the introduction provide sufficient background and include all relevant references?

(x)

( )

( )

( )

Are all the cited references relevant to the research?

(x)

( )

( )

( )

Is the research design appropriate?

( )

( )

(x)

( )

Are the methods adequately described?

( )

( )

(x)

( )

Are the results clearly presented?

( )

( )

(x)

( )

Are the conclusions supported by the results?

( )

( )

(x)

( )

Comments and Suggestions for Authors

Although the authors have made an effort to answer almost all the questions raised, the methodological errors have not been corrected, nor is the explanation given sufficient to understand why certain essential determinations have not been made to reach these conclusions. I am referring in particular to the fact that oxidative stress was measured in the blood and not in the supposedly affected tissue. In addition, I am surprised that the ethics committees accepted the collection of blood from the orbital sinus, a practice that is no longer used, and that this is not explained if this sample is obtained at the endpoint and there are other more appropriate methods.

Response:

Thank you for contemplating my efforts and suggested modifications to the manuscript. I have verified and rectified all methodological errors through track switching.

I respectfully disagree with the reviewers' position. I am specifically referring to the fact that oxidative stress was measured in the blood and not in the ostensibly affected tissue. I would like to inform you that the oxidative stress assay was performed on kidney tissue, not blood, and that the homogenization procedure and uses of the supernatant for the oxidative stress assay were highlighted in yellow in material and method sections 4.3 sample preparation.

I also disagree with the statement that retro-orbital procedures for animal blood withdrawal are not in use. If we need more blood for multiple hematological and biochemical tests, we can prefer retro orbital or decapitation procedure. Other procedures are effective for small amount of blood. In this investigation, blood was collected at the end of the experiment, and the animal was then sacrificed for the isolation of kidney tissue.

Submission Date

28 May 2023

Date of this review

15 Jun 2023 12:44:15

Round 3

Reviewer 2 Report (New Reviewer)

According to the authors, the changes/suggestions have been made, so I trust their veracity. This being the case, the article would be suitable for publication after a review of the writing since when modifying the text they have made some spelling errors.

For example, in the abstract "the second group received CFZ administration" redundant phrase. There is also a space left between CFZ and administration.

There are many spaces between words throughout the text while in other cases they have been removed. Review the writing.

My suggestion is that the writing of methods be reviewed to make it clear what type of sample is used in each experiment, as well as in the figure captions, since blood measurement alternates with tissue measurement and this makes their interpretation difficult.

Previously commented about the space and repetition of words

Author Response

Response to Reviewer-2 Round-3 Comments

Open Review

Quality of English Language

( ) I am not qualified to assess the quality of English in this paper.
( ) English very difficult to understand/incomprehensible
( ) Extensive editing of English language required
( ) Moderate editing of English language required
(x) Minor editing of English language required
( ) English language fine. No issues detected

Yes

Can be improved

Must be improved

Not applicable

Does the introduction provide sufficient background and include all relevant references?

(x)

( )

( )

( )

Are all the cited references relevant to the research?

(x)

( )

( )

( )

Is the research design appropriate?

( )

(x)

( )

( )

Are the methods adequately described?

( )

(x)

( )

( )

Are the results clearly presented?

( )

(x)

( )

( )

Are the conclusions supported by the results?

( )

(x)

( )

( )

Comments and Suggestions for Authors

According to the authors, the changes/suggestions have been made, so I trust their veracity. This being the case, the article would be suitable for publication after a review of the writing since when modifying the text they have made some spelling errors.

For example, in the abstract "the second group received CFZ administration" redundant phrase. There is also a space left between CFZ and administration.

Response: deleted the “administration” and space between CFZ and Administration

There are many spaces between words throughout the text while in other cases they have been removed. Review the writing.

Response: I try to remove the space between words throughout the manuscript.

My suggestion is that the writing of methods be reviewed to make it clear what type of sample is used in each experiment, as well as in the figure captions, since blood measurement alternates with tissue measurement and this makes their interpretation difficult.

Response: I appreciate your comments and clearly mentioned the blood used for the kidney function markers (Creatinine, Blood Urea Nitrogen, and Uric Acid), kidney tissue used for the inflammatory markers (TNFα and Caspase-3, IL-1β, IL-6 and Nrf2) as well as for histopathology.

Comments on the Quality of English Language

Previously commented about the space and repetition of words

Submission Date

28 May 2023

Date of this review  19 Jun 2023 10:01:14

This manuscript is a resubmission of an earlier submission. The following is a list of the peer review reports and author responses from that submission.

Round 1

Reviewer 1 Report

The manuscripts is an extension of the reported anti-inflammatory effects of thymoquinone (TQ) and provides no novelty on its mode of action. However, the provided data supports prior reports and could stimulate further research on the therapeutic use of TQ.

Other comments:

·         Page 2 of 15: Lines 2-5 are repetitive and could be consolidated.

·         Page 2 of 15, lines 10-11: The correct references should be provided. References 7 and 8 are not about CFZ.

·         Dosing design is currently confusing and should be properly described. Were CFZ rats given six ip 4mg/kg injections (24mg/kg) daily for 16 days? Were TQ1 and CFZ co-administered to animals daily for 16 days? A dosing scheme/figure would make the dosing regimen clearer.

·         Antioxidant assays (lipid peroxidation, glutathione, catalase and superoxide dismutase) assays should be briefly described in addition to the provided references.

·         Page 6 of 15: The last sentence should be corrected since TQ20 significantly increased GSH (Fig. 3A) compared to CNT.

·         Page 11 of 16: Please clarify what is meant by “TQ has an advantage over chemical medications in that it does not exhibit any signs of deadly toxicity.” TQ is a chemical, and every chemical has some dose dependent toxicity. Comment further on potential toxicity/side effects of TQ.

·         Any advantage of TQ over other antioxidant therapies reported to prevent drug-induced kidney injury?

·         Page 12 of 15 (Conclusion): In the 4th line from the bottom, lowering kidney “injury” markers rather than “function” markers is more accurate.  

·         Page 13 of 15: In the last sentence of the manuscript, it is unclear what is meant by “CFZ’s benefit when combined with other medications.” CFZ has been combined with dexamethasone without negative effects on its efficacy.

·         The author should comment on the anti-cancer or tumor promoting properties of TQ.

·         Does TQ have any effects on vascular activity (vasoconstriction/dilation) or is it only anti-inflammatory?

·         Tissue histology assessments to further support kidney injury would be useful to the readers.

Reviewer 2 Report

Generally, the topic is interesting and fits into the scope of the journal. However, some parts of the manuscript should be improved. For example, the Abstract should mainly contain the description of your own results, not general statements. The Results section should be more quantitative, i.e., it would be welcome to provide % of changes observed. Further, my most serious concerns focus on the number of animals (only five/per group) used in the study and on the statistical analysis of the data. Was the normal distribution test performed for statistical analysis?, Was the homogeneity of variances verified? How was the parametric test decision made? Moreover, the Authors should explain why did they choose these particular doses of TQ. In addition, the experiment on animals should be described in detail to avoid any guesses. For example, what kind of rats was used in the experiment - outbred or inbred Wistar rats? Was the animal behaviour throughout the experiment cheched? More remarks concerning the experiment alone are provided below. I also strongly recommend to include future perspective of the study - what is the next step? Besides, what are the limitations of the study? Besides, the points listed below need to be checked by the Authors.

 Abstract

1.     Line 29: ‘…antioxidant enzymes (GSH)….’. GSH is not an enzyme. 2.     Line 34: ‘…TNF alfa and caspase 3 expression’ (?) Both TNF alfa and caspase 3 were determined using the ELISA technique (page 2) Materials and methods

1.   Please provide the following information:

a)   cages and conditions prevailing therein:

-      in what cages (stainless steel or plastic cages,) the rats were placed during the experimental period; maybe the IVC system was used (?)

-      whether the rats were placed in the cages individually

-      the animal behaviour throughout the experiment

-      the mean initial weight of the rats in each Group

b)   the length of the adaptation period.

c)   Why was body weight not checked during the experimental period?, what was the reason?

d)   What kind of food/chow and water (distilled, tap, or deionised?) was provided to rats?

e)   5 rats per group. Why was the number of animals in each Group so small? what was the reason? Moreover, 15 rats seem to be missing - Line 101: ‘Randomly, forty male Wistar albino rats…’

2.   Was the serum used immediately for determination of biochemical parameters after collection?  

3.   Line 89, Table 1 does not show the PCR primers

4.   Line 124, suggests using the word ‘level’ instead of ‘concentration’

5.   Line 145 vs. Line 148, ‘TNF alfa’ vs. ‘interleukin’

6.   As for the abbreviations: According to the rules, each abbreviation should be explained when it is used for the first time and next the introduced abbreviation should be consequently used throughout the manuscript.

7.      Line 156, typo of ‘caspase’

8.      Line 162, the reviewer think that the Authors meant ‘Tukey-Kramer test’

9.      Line 163, six of five rats?

10.   Line 163, the reviewer think that the Authors meant ‘mean +/- SEM’

 Results

1.    ‘As compared to normal control’ - What does ‘normal control’ exactly mean?  Please check throughout the manuscript

2.   Line 181, the word ‘oxidant’ should be removed.

3.   Line 182, typo of ‘nmol’

4.   Lines 201-202, ‘In comparison,….significant results’ should be re-written. The grammar of the whole text of the manuscript should be thoroughly checked.

5.   Line 236, ‘Figures 6’ should be changed to ‘Figure 6’.

6.   Line 243, ‘We investigated….. CFZ injection’ – this is not a result.

7.   Figure 2, Y axis; ‘nanomole/gm tissue’ (?)

8.   Figures 3B and 3C (Y axis) / the text of the manuscript: ‘nmole’ vs. ‘nmol’ – it should be unified.

 Discussion

1.   Lines 305-307, ‘…the concentration of IL-1 beta, IL-6, TNF alfa, and caspase 3 in serum‘ (?) Only TNF alfa, and caspase 3 were determined in the serum.

 Conclusion

1.     Line 325, ‘..leading to ROS production’. ROS were not determined.

2.     Line 326, ‘expression of caspase 3’ (?)

3.     Lines 329-330, ‘expression of apoptotic genes’ (?)
